

# Variation in partner benefits in a shrimp—sea anemone symbiosis

C. Seabird McKeon[1] and James L. O'Donnell[2]

[1] Smithsonian Marine Station, National Museum of Natural History, Smithsonian Institution, Ft. Pierce, FL, USA

[2] School of Marine and Environmental Affairs, University of Washington, Seattle, WA, USA

## ABSTRACT

Symbiotic interactions, where two species occur in close physical proximity for the majority of the participants' lifespans, may constrain the fitness of one or both of the participants. Host choice could result in lineage divergence in symbionts if fitness benefits vary across the interaction with hosts. Symbiotic interactions are common in the marine environment, particularly in the most diverse marine ecosystems: coral reefs. However, the variation in symbiotic interactiines that may drive diversification is poorly understood in marine systems. We measured the fecundity of the symbiotic shrimp *Periclimenes yucatanicus* on two anemone hosts on coral reefs in Panama, and found that while fecundity varies among host species, this variation is explained largely by host size, not species. This suggests that shrimp on larger hosts may have higher fitness regardless of host species, which in turn could drive selection for host choice, a proposed driver of diversification in this group.

## INTRODUCTION

Variation in partner benefits has been proposed as an engine of diversification in symbiotic systems (*Schemske & Horvitz, 1984*; *Thompson, 1994*), and can have important effects on the survival of associated organisms (*McKeon & Moore, 2014*). The palaemonid shrimps of the subfamily potoniinae are one of the most diverse groups of reef animals (*Bauer, 2004*). Pontoniines are well known for their many and diverse symbiotic associations, on both Indo-Pacific and Caribbean reef systems (*Bruce, 1976*). Many shrimps of *Periclimenes* and related genera can be found as commensals on hosts as varied as file clams, sponges, and echinoderms as well as cnidarian hosts (*Li, 1993*). Several species are found in association with sea anemones (Actiniaria) and reef corals. The diversity of the pontoniines, and frequent symbiotic interactions with other taxa has spurred interest in the role of symbioses in diversification of the subfamily. Molecular studies have identified potential evolutionary groups and pathways to symbiosis with different marine organisms (*Kou et al., 2015*), but little work has been done on the ecological forces that might be at the root of such divergence.

The ecological interactions between pontoniine shrimps and their hosts may be complex, incorporating feeding strategy, nitrogen transfer, as well as defense and secondary mutualisms, such as cleaning behaviors (*Limbaugh, Pederson & Chace, 1961*; *Spotte, 1996*). Among the best studied of the pontoniines, *Periclimenes yucatanicus*

Corresponding author
C. Seabird McKeon, mckeons@si.edu

(Ives, 1891), the 'Spotted Anemone Shrimp,' is a common exosymbiont of sea anemones in shallow marine habitats from the southern United States to the Caribbean coast of South America (reviewed in *Spotte et al. (1991)*). Notable differences in coloration and other morphological features exist across this taxon's range (*Spotte et al., 1991*; *Wicksten, 1995*). *P. yucatanicus*, as with several other Caribbean pontoniines, maintains host associations with many species of sea anemones and corallimorpharians across a broad range of sizes and morphologies. Interactions of anemone shrimps with this broad range of hosts has been examined in the context of rapid environmental change (*Silbiger & Childress, 2008*) and host choice behavior (*Mascaro et al., 2012*). However, little work has been done to investigate the question of differential fecundity and fitness resulting from host variation (*Ross, 1983*), and the implications of this on diversification in symbiotic lineages. This study investigated differences in size and fecundity of *P. yucatanicus* on two different anemone host species: the Sun Anemone *Stichodactyla helianthus* (Ellis, 1767), and the Corkscrew Anemone *Bartholomea annulata* (Lesueur, 1817).

## METHODS

We conducted our study in protected waters close to the Smithsonian Tropical Research Institution field station, Bocas del Toro, Panama, where *P. yucatanicus* are abundant, readily visible at close range, and easily captured. They are often found in groups on a single sea anemone host, with larger individuals bearing eggs externally, leading to the belief that the species is sequentially hermaphroditic. While *P. yucatanicus* can be found on a wide variety of sea anemones and anemone-like hosts in the Caribbean, we focused on their interactions with the two most common host species in the area: *Stichodactyla helianthus* and *Bartholomea annulata*. These two sea anemone host species are of different architecture: *Bartholomea* has a small oral disc with long, thin tentacles, while *Stichodactyla* has a very broad oral disc with short stubby tentacles.

Shrimp and anemones were surveyed and/or collected from shallow reefs (<2 m) using snorkel equipment within a mixed seagrass/patch reef/mangrove habitat matrix. The tentacular spread of the anemone was carefully measured avoiding the disturbance and retraction of the anemone. The longest straight-line distance between the tips of two tentacles based at opposite sites of the oral disc was measured to the nearest centimeter (cm) using a tape measure as a proxy of body size for both taxa of sea anemone. Other measures of sea anemone size (disc diameter, water displacement) were unsuccessful. All *P. yucatanicus* on an individual sea anemone were counted prior to disturbance, collected using a dip net or slurp gun, and returned to the lab in individual plastic bags. Shrimp were never removed from the water.

The carapace length of each shrimp was measured with calipers to the nearest half millimeter (mm) as a proxy of body size, and examined for the presence or absence of eggs on the pleopods. As eggs of the shrimps in this genus grow in size, and clutches are reduced in number with growth (*Azofeifa-Solano, Elizondo-Coto & Wehrtmann, 2014*), clutches of older eggs with visible eyespots and near hatching were not included in the analysis. Eggs were irrigated off of the pleopods of gravid shrimp using a syringe, and counted using a

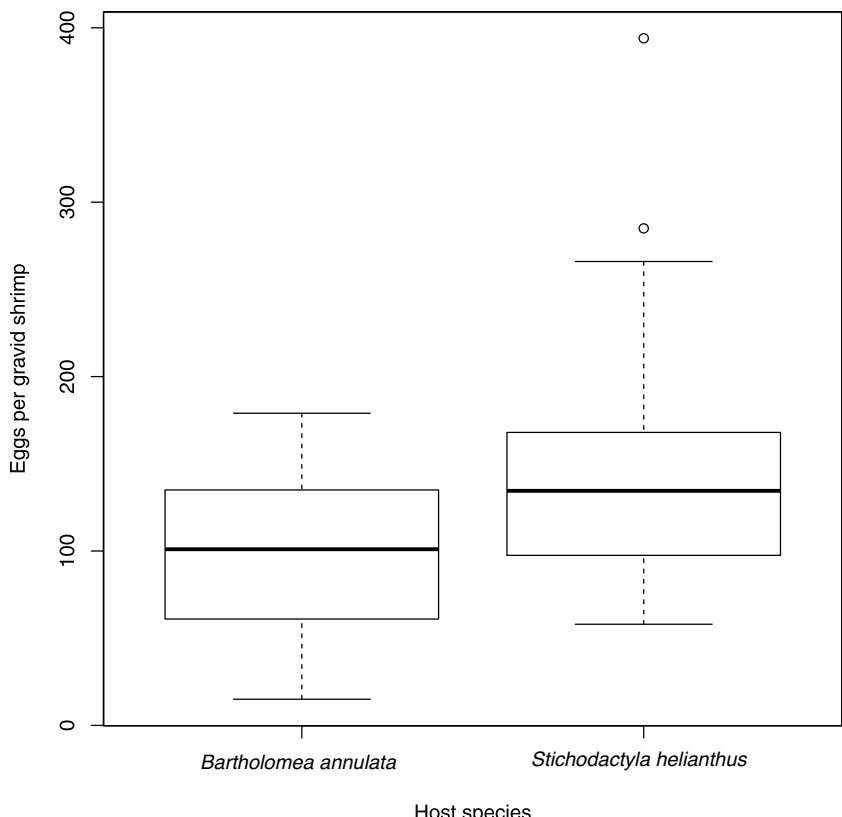

**Figure 1 Number of eggs per gravid individual of *Periclimenes yucatanicus* found on each of two host sea anemone species.** Central lines represents the median value, with boxes encompassing the interquartile range. Whiskers extend to the more central value of either the furthest data point or 1.5 times the interquartile range, and outliers are represented as dots.

dissecting microscope and a hand counter. The day after collection, shrimp were returned alive to the site from which they were gathered. Sampling continued until 30 or more egg-bearing individuals were collected from each host species.

We tested for the effect of host sea anemone species on fecundity of *P. yucatanicus* while controlling for the effect of shrimp body size using an analysis of covariance (ANCOVA). All statistics were calculated in the statistical programing environment R (*R Core Team, 2015*).

## RESULTS

A total of 32 gravid shrimp with eggs of appropriate size were gathered from *S. helianthus*, 34 were collected from *B. annulata*. The total number of individual shrimp, both gravid and not gravid, per individual host did not differ between the two hosts (*B. annulata* $m = 1.35$ $sd = 0.69$; *S. helianthus* $m = 1.81$ $sd = 1.38$; Welch's two sample $t$ (45.04) $= -1.70$, $p = 0.1$). In the samples gathered, only one shrimp on a sea anemone was gravid. Shrimp individuals carried a greater number of eggs on *S. helianthus* ($m = 145.03$, $sd = 68.98$) than *B. annulata* ($m = 98.97$, $sd = 42.35$; Welch two sample $t$ (50.88) $= 3.245$, $p = 0.001$; Fig. 1). As expected of decapod crustaceans (*Corey & Reid, 1991*), there was a positive linear
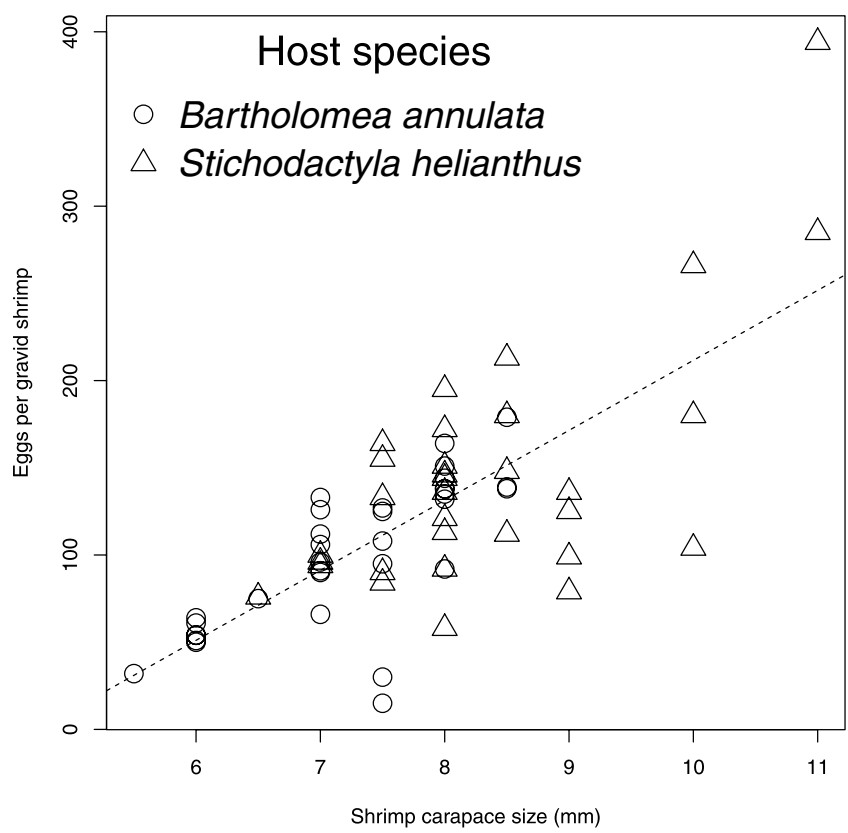

**Figure 2 Number of eggs per gravid individual of *Periclimenes yucatanicus* plotted against shrimp body size.** The species of host from which each individual was collected is indicated by the marker shape (circles, *Bartholomea annulata*; triangles, *Stichodactyla helianthus*), and the dashed line represents the relationship given by the linear regression ($F(1, 64) = 81.71$, $p < 0.0001$, $R^2 = 0.5608$).

relationship between the number of eggs carried by an individual shrimp and its carapace length ($F(1, 64) = 81.71$, $p < 0.0001$, $R^2 = 0.5608$, Fig. 2). Sea anemones of the species *S. helianthus* ($m = 16.03$ cm, $sd = 4.30$) were larger than those of *B. annulata* ($m = 13.56$ cm, $sd = 5.02$; Welch two sample $t$ (63.46) $= 2.15$, $p = 0.035$), and shrimp were larger on *S. helianthus* ($m = 8.34$, $sd = 1.10$) than on *B. annulata* ($m = 7.19$, $sd = 0.862$; Welch's two sample $t$ (58.66) $= 4.71$, $p < 0.0001$), but we found no effect of anemone species on shrimp size when accounting for sea anemone size ($F(1, 62) = 0.0001$, $p = 0.993$). We incorporated these variables into an ANCOVA model to understand the effect of sea anemone species, shrimp size, and their interaction effects on shrimp egg number. There was no effect of the interaction between anemone species and shrimp size on egg number ($F(1, 62) = 0.22$, $p = 0.64$), indicating that the host sea anemone species had no effect on the slope of the relationship between shrimp size and egg size. A reduced, additive linear model (egg number ∼host species + shrimp size) found no effect of sea anemone species on egg number ($t = -0.023$, $p = 0.982$), but a significant effect of shrimp size ($t = 7.725$, $p < 0.0001$), resulting in an overall positive effect of shrimp size on egg fecundity, independent of host anemone species ($F(2, 63) = 40.22$, $p < 0.0001$, $R^2 = 0.547$). Finally, there was a significant combined effect of shrimp size and anemone size on egg number,

irrespective of host sea anemone species $F(3, 62) = 39.89$; $p < 0.0001$; $R^2_{adj} = 0.6422$. Interaction between anemone size and species suggest that the pattern observed with *P. yucatanicus* is driven largely by host size, with larger sized hosts capable of supporting larger and more fecund shrimp.

## DISCUSSION

Considerable theoretical and empirical research has focused on tradeoffs and correlations among critical components of reproductive effort, such as age and size at first reproduction, schedule of reproduction, egg size and clutch size (*Stearns, 1992*). Such focus has rarely been applied to symbiotic taxa, and even less frequently to marine symbiotic taxa. In many taxa, there is a positive correlation between clutch size or ovarian egg number and body length (*Corey & Reid, 1991*; *Salthe, 1969*), as seen in *P. yucatanicus*. The differences in clutch size seen in shrimp on the two host species represent a facet of life history theory of possible evolutionary importance.

As discussed by *Mascaro et al. (2012)*, previous efforts to examine host choice in pontoniine shrimps (*Guo, Hwang & Fautin, 1996*; *Silbiger & Childress, 2008*; *Spotte et al., 1991*) have yielded complex results demonstrating across and within taxon variation. Studies of the ecological interactions of anemone shrimps with hexacorallian hosts have included studies of distribution on and among hosts, and space-partitioning amongst co-occuring symbiotic organisms (*Bos & Hoeksema, 2015*; *Hayes & Trimm, 2008*; *Hoeksema & Fransen, 2011*; *Khan et al., 2003*; *Mascaro et al., 2012*). *Stanton (1977)* noted that anemones with larger fleshy tentacles were more likely to host commensal crustaceans. The work of *Silbiger & Childress (2008)* demonstrated that *P. yucatanicus* in Florida strongly preferred a third species of sea anemone, *Condylactis gigantea*, over the two species presented in this work. *C. gigantea* has fleshy tentacles and is frequently larger than *S. helianthus* and *B. annulata*.

The work presented here suggests host preference may be determined by host size, via its effect on fecundity in *P. yucatanicus*. Shrimps of this species are capable of attaining larger sizes and greater fecundity in larger host sea anemones. Increased fecundity of *P. yucatanicus* on the larger *Stichodactyla* hosts may imply a fitness benefit to host selection. The relationship between host size and shrimp size may be further mitigated by the presence of additional symbionts- of the same or different species (*McKeon & Moore, 2014*). These complex relationships between fecundity and community may be productive for future research efforts.

In the absence of host specificity, can differential reproductive success among hosts help to the explain patterns of diversification that we observe in pontoniine shrimps? Social (intra-specific) competition for resources other than mates has been invoked as a driver of speciation (*West-Eberhard, 1983*), though most experimental examples of differential fecundity come from highly specific relationships (*De Marisco & Reboreda, 2008*). Indeed, it remains unclear whether host specificity is a basal or derived trait (*Lanyon, 1992*; *Rothstein, Patten & Fleischer, 2002*). The work of *Kou et al. (2015)* on the systematic arrangement of host preference in Pacific pontoniine genera provides an intriguing hypothesis that

specialization in actinarian (sea anemone) hosts may be representative of a intermediate placement in the pattern of divergence of the group, but does not address the role of host specificity for individual shrimp taxa. While the variation in host-shrimp partnerships and resultant differences in fecundity meet the 'precondition for mutualism specialization' proposed by *Schemske & Horvitz (1984)*, further information on host specificity and the placement of the Atlantic pontoniine fauna within an evolutionary context is needed to see if the Atlantic fauna follows a similar route to divergence and specialization as the Pacific fauna.

## ACKNOWLEDGEMENTS

Editorial comments and support from Candy Feller and Kyle Summers were critical to the success of this work, as was the friendship of Tom Oliver and Rita Steyn during one crazy summer in Panama. Magnus Johnson, Bert Hoeksema, and an anonymous reviewer provided feedback that greatly improved the quality of this manuscript. This is Smithsonian Marine Station contribution #1016.

### Funding

We received support from the Smithsonian Tropical Research Institute and the Organization for Tropical Studies through the STRI/OTS scholarships and fellowships. Travel funding was received from East Carolina University and The Rivers Foundation. The funders had no role in study design, data collection and analysis, decision to publish, or preparation of the manuscript.

### Grant Disclosures

The following grant information was disclosed by the authors:
STRI/OTS scholarships and fellowships.
East Carolina University.
The Rivers Foundation.

### Competing Interests

C. Seabird McKeon is an Academic Editor for PeerJ. The authors declare there are no competing interests.

### Author Contributions

- C. Seabird McKeon conceived and designed the experiments, performed the experiments, analyzed the data, contributed reagents/materials/analysis tools, wrote the paper, prepared figures and/or tables, reviewed drafts of the paper.
- James L. O'Donnell analyzed the data, contributed reagents/materials/analysis tools, wrote the paper, prepared figures and/or tables, reviewed drafts of the paper.

### Supplemental Information

Supplemental information for this article can be found online at http://dx.doi.org/10.7717/peerj.1409#supplemental-information.

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
