# Peer review of "Variation in partner benefits in a shrimp—sea anemone symbiosis"

_PeerJ, doi:10.7717/peerj.1409_

## Round 0.1 · original submission · Major Revisions

Both reviewers agree that this manuscript requires major revision but have made some very useful comments that should help you bring it up to a good standard.

I suggest that you take a look at the likelihood that there is a combined effect of carapace length and anemone size (irrespective of species) on egg number, i.e. eggno = cl * anemsize.

·

Basic reporting

Background: The authors may consider comparisons with other studies on host selection and spatial distribution of commensal shrimps in sea anemones (e.g. Khan et al. 2003, 2004; Hayes & Trimm 2008; Mascaro et al. 2012) or in scleractinians resembling sea anemones (Hoeksema & Fransen 2011; Bos & Hoeksema 2015).
Figures should indicate the latin names of the species concerned.

Experimental design

Sampling: If the sea anemones and shrimps were collected by snorkeling down to 2 m depth, could any of the shrimps have escaped? This does not appear to be a very secure method. The authors should mention how the animals were collected (jars, plastic bags). Sea anemones and their associated fauna may be difficult to catch when they are (partially) situated inside crevices or underneath rocks. Were such animals ignored?

Validity of the findings

The authors mention that the shrimps may occur in groups but I could not find information on the number of shrimp individuals per host. The ms only mentions females but no males. Were there any other animals with which the host had to be shared? If the study concerns host size, then the presence of co-inhabitants may be relevant with regard to space partitioning. See references mentioned above.

Additional comments

The relevance of several statements in the paper (particularly in the discussion) is not clear to me. I have given additional comments on the pdf.

Reviewer 2 ·

Basic reporting

The aim of the study is clearly stated and is directed to explore differences in size and fecundity of P. yucatanicus that could be related to host species in order to find evidence of patterns in the association of this shrimp and its host anemones. The authors discuss their results in the context of the diversification observed in Pontoniine shrimps. I suggest, however that some ideas that were very briefly posed in the last paragraph, should be more fully explored for the benefit of a richer and more productive discussion.
1) A more elaborate insight of the way the authors view their results in the context of these concepts will surely encourage robust and innovative hypotheses to be formulated in future research.
• host specificity as a basal or derived trait (Rothstein et al. 2002).
• intermediate placement in the pattern of divergence of the group (Kou et al., 2015).
• the variation in host-shrimp partnerships and resultant differences in fecundity meet the ‘precondition for mutualism specialization’ (Schemske & Horvitz,1984).

2) Figures definitely have insufficient information:
• titles in axes have no units; the reader needs to go back to the text to find what abbreviations “cork” and “sun” stand for;
• it is clear that shrimp size was measured to the nearest 0.5 (cm, mm?) but there is no information as to what units and how this was measured either in the legend hear or in the text (Vernier calliper?)
• dot size is a bit too small, and could be made larger (in ggplot this is easy)

3) Legends for Figures are insufficient to adequately interpret graphical results and do not “stand on their own”. Readers need to refer to the text to correctly understand figures:
• Figure 2 shows a boxplot, but no information is given as to what box limits and whiskers represent (mean or median, sd, se, CI, Chambers’ standard features?);
• it is not clear if the number of eggs has been made relative to individual shrimp (number of eggs per shrimp) or to individual anemone (with no regard as to how many shrimp were found on that particular host. This information is crucial to understand the numerical and graphical output of the statistical analysis and its interpretation.

Experimental design

Whilst the authors clearly defined the research question, and the methods used are generally consistent with the aims stated, there in much information missing from the materials and methods in order for the study to be repeatable. In the materials and methods section:
1) There is ambiguity as to how procedures to register the size of shrimp and anemones were carried out: how did authors proceed to ensure a precise measure of the “tentacular spread…avoiding the disturbance and retraction of the anemone”, particularly in the field.
2) Why were older eggs with visible eyespots not included in the analysis?
3) There is no information as to how many host anemones had to be sampled for a sample size of 30 “or more” (how many more?) egg-bearing individuals to be collected. No information of the n used in the comparison of anemone size between species.
4) There is no information on the mean number of eggs per individual shrimp or a measure of the variability associated. This impedes the direct comparison of a measure of “standardised” fecundity (relative to the individual) between host species, both in terms of a central tendency and a dispersion parameter. If the correlation between fecundity vs shrimp size was estimated in this way, this is not clarified in the methods section and the reader needs to deduce it.

Validity of the findings

I found the results presented by the authors generally convincing, but I believe some work is still needed in the results section, particularly with the statisitical analysis and its numerical and graphical outputs:
1) Information as to how many gravid shrimp per individual host were found in average (± sd) is needed, since this will give an idea of how constant the occurrence of gravid shrimp is relative to host species. If many gravid shrimp were found in any single host, then clarification as to how many and how were shrimp selected in each case is required (the spread sheet has a column heading with the legend NUMBERGROUP, and it is unclear whether it refers to the number of shrimp or some other feature).
2) The authors state that the relationship between shrimp size and egg number is linear. Although this could be the case, this is not clear from Figure 3, and relatively high values of R2 does not warrant linearity. A quick visualisation of the residuals of the regression might help ensure the linearity of the relation and provide the needed proof to validate the additive model used in the statistical procedure. A good reference for visual analysis of residuals in regression is Montgomery,et al., 2006 (Introduction to linear regression analysis) and Zuur et al., 2007 (Analysing Ecological Data Series: Statistics for Biology and Health)
3) When applying an ANCOVAR, the authors disregard the fact that egg number is a discrete response variable, which most probably does not conform to a normal distribution. If a visual analysis of residuals provides proofs of an increased variance in egg number as anemone (and as shrimp for that matter) increase in size (heterogeneity of variance associated to the continuous explanatory variable), then a transformation of the number of eggs could improve the model. Even if such transformation does not modify the general statistical result, the validation of the model by looking at the residuals is now a standard procedure.
4) The authors explain they used a reduced model to find “no effect of anemone species on egg number, …. a significant effect of shrimp size, ….. resulting in an overall positive effect of shrimp size on egg fecundity, independent of host anemone species”. If I understand correctly, this is a procedure to reduce the statistical model to only the explanatory variables that significantly explain the response. The order in which the main terms are assessed is required, since it is well known that this may have an a effect on the significance of the variable left in the model, particularly when there is suspicion of high correlation between explanatory variable, which appears to be the case (anemone size and host are correlated).
5) Finally, Figure 3 shows 2 regression lines, one for each host species. These are parallel because they are the depicting the result of a non-significant interaction term in the ANCOVAR. However, no differences were found between anemones, so it would follow that those two regression lines, are statistically only one. If this is the case and the authors feel that both lines are required to stress their results, I would still recommend this be explained at some point in the text or figure legends.

Additional comments

The contribution of this original study is highly important, not only because it provides of new insights as to where to look for the ecological mechanisms underlying the patterns of divergence in this group, but because it also sheds light on the meaning of the relationship (function) between host and shrimp, of what is given and taken and what advantages (or not) the trade-of represents to the individual.

---

## Round 0.2 · Minor Revisions

There are just a couple of minor points made by the reviewer that are never-the-less worthy of attention. If you can deal with these I will recommend that the article is accepted on the next iteration.

Reviewer 2 ·

Basic reporting

Authors have complied with all suggestions both regarding the information in the Results and Discussion sections, and ms has been much improved.

Experimental design

The materials and methods are much clearer, and can now be be repeated by another research group. I still found it a bit confusing to have these two sentences one after the other. For somebody unacquainted with the methods, a small change might make it easier to read through and understand (I had to reaad it again a couple of times to understand that the first sentence is regarding all individual shrimp, gravid or not).

Lines 94-96: The number of individual shrimp per individual host did not differ between the two hosts (B. annulata m = 1.35 sd = 0.69; S. helianthus m = 1.81 sd = 1.38; Welch’s two sample t(45.04) = -1.70, p = 0.1). In the samples gathered, only one shrimp on a sea anemone was gravid.

Validity of the findings

In my first review and now again, I found the authors present convincing results to support that the number of eggs in P. yucatanicus is not determined by the species of host anemone, but by its size; and that this relation is the natural result of another two direct relations: number of eggs increases with shrimp size, and shrimp size increases with anemone size. The reason for this is that judging by the p, F and t values given by the authors (there are no marginally significant results), I am quite sure the overall statistical outcomes of the tests and the authors’ conclusions will be the same using these or another more sophisticated techniques. I believe in the easier, the better.
My questions were regarding the quality of the fitted model, which in turn provides confidence in both test results (reject of fail to reject the null), and more importantly, in the estimation of the coefficients in the model (intercepts and slopes). Given that a graphic analysis of the residuals is a standard procedure in statistical methods applied in the field of ecology, I think the authors should present those graphical visualisations as part of the supplementary material. It would provide much more information than just the data and the R code, and will allow the reader to decide on his own how far is the actual model from the desired approximation to normality, homogeneity, etc.

---

## Round 0.3 · accepted · Accept

Thank you for your rapid response to the last revision. I hope the paper is well received.